# Estimation of Respiratory Rate during Biking with a Single Sensor Functional Near-Infrared Spectroscopy (fNIRS) System

**DOI:** 10.3390/s23073632

**Published:** 2023-03-31

**Authors:** Mohammad Shahbakhti, Naser Hakimi, Jörn M. Horschig, Marianne Floor-Westerdijk, Jurgen Claassen, Willy N. J. M. Colier

**Affiliations:** 1Artinis Medical Systems, B.V., Einsteinweg 17, 6662 PW Elst, The Netherlands; 2Biomedical Engineering Institute, Kaunas University of Technology, K. Barsausko 59, LT-51423 Kaunas, Lithuania; 3Department of Neonatology, Wilhelmina Children’s Hospital, University Medical Center Utrecht, Lundlaan 6, 3584 EA Utrecht, The Netherlands; 4Donders Institute for Brain, Cognition and Behaviour, Radboud University, Houtlaan 4, 6525 XZ Nijmegen, The Netherlands

**Keywords:** fNIRS, respiratory rate, physical activity, wearable

## Abstract

Objective: The employment of wearable systems for continuous monitoring of vital signs is increasing. However, due to substantial susceptibility of conventional bio-signals recorded by wearable systems to motion artifacts, estimation of the respiratory rate (RR) during physical activities is a challenging task. Alternatively, functional Near-Infrared Spectroscopy (fNIRS) can be used, which has been proven less vulnerable to the subject’s movements. This paper proposes a fusion-based method for estimating RR during bicycling from fNIRS signals recorded by a wearable system. Methods: Firstly, five respiratory modulations are extracted, based on amplitude, frequency, and intensity of the oxygenated hemoglobin concentration ($O_{2}Hb$) signal. Secondly, the dominant frequency of each modulation is computed using the fast Fourier transform. Finally, dominant frequencies of all modulations are fused, based on averaging, to estimate RR. The performance of the proposed method was validated on 22 young healthy subjects, whose respiratory and fNIRS signals were simultaneously recorded during a bicycling task, and compared against a zero delay Fourier domain band-pass filter. Results: The comparison between results obtained by the proposed method and band-pass filtering indicated the superiority of the former, with a lower mean absolute error (3.66 vs. 11.06 breaths per minute, p<0.05). The proposed fusion strategy also outperformed RR estimations based on the analysis of individual modulation. Significance: This study orients towards the practical limitations of traditional bio-signals for RR estimation during physical activities.

## 1. Introduction

The advent of wearable systems for the measuring of vital signs has started a revolution in non-laborious healthcare and sports science practices [1]. Unarguably, such systems have provided the possibility of customized continuous monitoring of physiological parameters, such as heart rate (HR), blood pressure, and respiratory rate (RR), which are of great value for both clinical and research purposes [2,3].

A majority of wearable systems indirectly estimate the aforementioned parameters using bio-signals, such as the photoplethysmography (PPG) and electrocardiography (ECG) [4]. Despite the excellent performance during resting state, wearable systems usually do not show a satisfactory performance when the subject is engaging in physical activity [5]. The reason for such failure is the substantial susceptibility of bio-signals to motion artifacts originating from a subject’s movements, which are inevitable during physical activity. Amongst the physiological parameters mentioned, estimation of RR from bio-signals contaminated with motion artifacts is the most challenging [6].

Alternatively, functional Near-Infrared Spectroscopy (fNIRS) can be employed and is less sensitive to a subject’s movements. Specifically, fNIRS is an optical brain imaging technique that quantifies oxygenated ($O_{2}Hb$) and deoxygenated (HHb) hemoglobin concentrations, related to the neural activities in the cerebral cortex. Not only does fNIRS provide a better spatial resolution than electroencephalography (EEG), it also outperforms functional magnetic resonance imaging (fMRI) in terms of temporal resolution [7]. More importantly, compared to other bio-signal monitoring techniques, fNIRS is more motion tolerant, i.e., the subject’s movements do not necessarily affect the quality of the data [8,9]. Due to its noninvasive and mobile nature, fNIRS also has the advantage of being usable in outdoor applications [10]. Indeed, the advent of lightweight single sensor fNIRS equipment has opened new doors for outdoor studies, and, in particular, fitness applications [11,12,13,14].

Opportunely, fNIRS is contaminated by non-cerebral interference cited from physiological systemic activities, such as HR, blood pressure, Mayer waves, and respiration [15], providing the possibility of extraction of complementary physiological parameters. Although the HR estimation from fNIRS measurements has been addressed in several studies, no research, except for our previous study [16], has yet investigated the possibility of RR estimation from fNIRS. Respiration is one of the main physiological interferences in fNIRS, manifesting in its spectrum (∼0.2 to 0.4 Hz) [17,18]. The reason for such contamination is twofold; changes of the blood flow within the body during inhaling and exhaling, and the influence of respiratory oscillations on cerebral blood volume and flow [19]. Therefore, it can be presumed that RR changes are disclosed in the fNIRS signals.

Due to the similar principles of PPG and fNIRS measurements, and, in particular, $O_{2}Hb$ signal, and considering the fact that cerebral blood volume and flow changes are proportional to fNIRS output [20], it can be expected that respiratory modulations exhibited in PPG behave similarly in $O_{2}Hb$ signals. Therefore, we hypothesize that respiration can influence the $O_{2}Hb$ signal in three aspects; frequency, intensity and amplitude.

In our previous research, [16], we showed the possibility of RR estimation from an $O_{2}Hb$ signal in a resting state. However, the main challenge is to estimate RR when subjects are performing physical activities. The objective of this paper is, thus, to propose a new method for estimation of RR from $O_{2}Hb$ signals during a bicycling task. The basis of the proposed method involves the following: (i) extraction of five respiratory modulations, based on amplitude, frequency, and intensity of an $O_{2}Hb$ signal, (ii) computation of the dominant frequency of each modulation, and (iii) application of a mean-based fusion strategy on the computed dominant frequencies to estimate RR.

## 2. Proposed Method

The block diagram of the proposed method is shown in Figure 1. As displayed, it mainly consists of two blocks; pre-processing and RR estimation.

### 2.1. Pre-Processing

The first step of the proposed method is to find a high quality fNIRS channel for the analysis. Towards this aim, firstly, the modified Beer–Lambert law is employed to convert raw optical density signals to $O_{2}Hb$ and HHb changes in concentrations as follows [21]:(1)ΔOD(λ)=ϵ(λ)·ΔC·L·DPF
where OD, λ, ϵ, *C*, *L*, and DPF stand for optical density, wavelength, absorption coefficient, concentration, physical path length, and differential path length factors, respectively. Then, the signal quality index (SQI) algorithm [22] is used, which numerically scores the signals from 1 (low) to 5 (high) quality. After finding the fNIRS channel with the highest signal quality, the corresponding $O_{2}Hb$ signal (Figure 2a) is filtered by a wavelet transform [23] for elimination of possible motion artifacts (Figure 2b). Then, an FIR low-pass filter with cut-off frequency at 4 Hz is used to remove high frequency noise originating from non-physiological sources (Figure 2c).

### 2.2. RR Estimation from $O_{2}Hb$

#### 2.2.1. Background

As was already mentioned, the $O_{2}Hb$ signal originating from fNIRS functions on the same basis as PPG, and, thus, respiratory modulations in PPG can also appear in the $O_{2}Hb$ signal. It is common knowledge that HR increases during inhalation and decreases during exhalation, and, thus, given the fact that each cycle of the $O_{2}Hb$ signal is a heart beat, respiration causes frequency modulation in the $O_{2}Hb$ signal [24]. On the one hand, increasing and decreasing of the intrathoracic pressure during inhalations and exhalations leads to a variation in blood volume, known as baseline wander, and, thus, respiration causes intensity modulation in the $O_{2}Hb$ signal [25]. On the other hand, reduction of the blood stroke volume during inhalation, because of alternation in intrathoracic pressure, leads to changes in the $O_{2}Hb$ signal’s strength, causing respiratory amplitude modulation [26].

#### 2.2.2. Extraction of Respiratory Modulations

In this paper, two frequency and two intensity respiratory modulations were extracted from the $O_{2}Hb$ signal, along with one amplitude respiratory modulation, for RR estimation. Regarding the frequency modulations, the following were computed: (i) the distance between two consecutive peaks and (ii) the distance between troughs and peaks. For the intensity modulations, two baseline wanders were formed: (i) based on peaks and (ii) based on troughs. Regarding the amplitude modulation, the absolute difference between the magnitude of troughs and peaks were computed.

#### 2.2.3. Trough and Peak Detection

In order to localize peaks and troughs, we used our previous method for trough detection from the $O_{2}Hb$ signal, wherein the $O_{2}Hb$ signal is first normalized between −1 and 1, and, then, local minima, with values lower than average of the $O_{2}Hb$ signal, are considered as potential troughs. Yet, the existence of motion artifacts may lead to a fallacious analysis. To overcome this issue, the corresponding magnitude of potential troughs were set on a vector, e.g., z[n], and, then, troughs with a lower value than Mean(z[n])+3×std(z[n]) were discarded [16]. Afterwards, the maxima between two consecutive troughs were considered the peaks.

#### 2.2.4. Processing of the Extracted Modulations

After forming the extracted modulations, up-sampling is an essential step, as the modulations were formed based on irregular sampling, whereas regularly sampled signals are often required for the subsequent analysis [27]. To this end, the cubic spline interpolation method was used, which has already been shown to be promising in PPG studies [28].

Before computing the dominant frequency of the up-sampled extracted modulations, it is also necessary to remove non-respiratory fluctuations, e.g., Mayer waves [29], which can hamper the accurate detection of the corresponding RR frequency [27]. Non-respiratory fluctuations can have both very low and very high frequency components. Unfortunately, there is no assent on the ideal frequency range of respiration. Indeed, such a range might need to be regulated based on the subject’s background, e.g., age [30]. In our previous study, it was shown that low-frequency components of non-respiratory fluctuations can be removed by a 3s moving average filter. For the maximum frequency of RR, we adopted a threshold based on the HR [31] as follows:(2)$$F_{Max}=A\times \frac{HR}{60},$$
where *A* is a constant. It is worth mentioning that HR was estimated based on the number of detected peaks on a 50s window length. The regulation of the constant *A* is described in Section 4.

After applying the moving average filter on the up-sampled modulations, the fast Fourier transform (FFT) algorithm was computed and the dominant frequency in the range of $\left[0\;\;F_{Max}\right]$ considered to be the RR.

#### 2.2.5. Mean-Based Fusion Method for RR Estimation

According to several studies, there is no optimal modulation for RR estimation [32]. For example, it was shown that frequency modulations may not be efficient for elderly subjects [33]. Therefore, computing the RR based on a fusion strategy, where several modulations are combined, can be more effective. Here, we used a simple mean fusion method where the final RR is computed based on the average of computed dominant frequencies of all modulations.

### 2.3. Method under Comparison

To the best of our knowledge, there is no direct method to estimate RR from fNIRS. Yet, some studies tried to portion out respiratory components from fNIRS measurements [19,34]. In this paper, we compared performance of the proposed method to a band-pass filtering strategy to extract respiratory components from the $O_{2}Hb$ signal. As described in [19], a zero delay Fourier domain band-pass filter, with cut-off frequencies from 0.2 to 0.6 Hz, was used. In this paper, after finding the fNIRS channel with the highest signal quality, the corresponding $O_{2}Hb$ signal was filtered and the dominant frequency within the range [0.2 to 0.6] Hz considered to be the estimated RR.

### 2.4. Performance Index

In order to assess the performance of RR estimation using both methods, the absolute error (AE) between the reference and estimated RRs was computed as follows:(3)$$\mathrm{AE}=|{\mathrm{RR}}_{ref}-{\mathrm{RR}}_{est}|,$$

## 3. Data Collection

### 3.1. The Experimental Setup

The experimental setup involved an ergometer bike, a wireless single-sensor fNIRS-system (PortaLite MKII, Artinis Medical Systems B.V., the Netherlands), placed on the left hemispheres of the prefrontal cortex, and a respiratory chest belt connected to a TMSi SAGA 32+/64+ amplifier (Twente Medical Systems International B.V., The Netherlands), with a sampling rate of 4000 Hz (Figure 3a). The fNIRS system was equipped with three long channels (transmitter–receiver distance up to 41 mm), recording data with nominal wavelengths of 760 and 850 nm, and a sampling rate of 100 Hz.

### 3.2. The Experimental Paradigm for Data Recording

The data recording procedure is displayed in Figure 3b. It comprised of a 60 s resting period block, followed by two bicycling blocks detached by a 60 s resting period, and the final resting period block lasting for 60 s. At each bicycling block, subjects bicycled for 8 min, wherein every minute the resistance of the ergometer increased by 1. In the first bicycling block, subjects were allowed to regulate their bicycling speed arbitrarily. In the second bicycling block, the subjects were asked to keep the speed constant around 18–22 km/h. In the whole procedure, subjects were allowed to freely move their heads and arms.

### 3.3. Participants

Twenty-two young healthy subjects with varied ethnicity and fitness endurance volunteered to take part in the experiment (Table 1). None of the subjects had any psychological and neurological disorders nor were they under any specific medications in the week of the experiment. Before starting the experiment, all participants were briefed about the study and signed a consent letter. A waiver for formal ethical approval was obtained from the medical ethics committee (CMO Arnhem-Nijmegen) with the ethical code of 2021-8284. All regulations and guidelines needed by the Declaration of Helsinki were followed. The data were collected at the premises of Artinis Medical Systems B.V., Elst, The Netherlands.

## 4. Results

In this section, the experimental results obtained from the proposed method and the method under comparison are described. Before starting the analysis, the respiratory reference signals were manually inspected and trials with an unreliable RR were rejected. After the manual inspection, the data associated with subject 21 were completely discarded. It should be noted that after discarding unreliable respiratory signals, 418 trails with a length of 50 s were used for the analysis.

### 4.1. Regulation of RR Maximum Frequency

The only parameter of the proposed method that was required to be tuned was the constant *A*, which regulated the RR maximum frequency based on HR. For this purpose, 78 trials, having 50 s lengths, from five subjects were randomly selected and values from 0.2 to 0.4 with a step size of 0.05 were investigated. To find the best fit, the lowest mean AE between the reference and estimated RR was considered (Figure 4). Although there was no significant difference between the obtained results by A=0.3 and A=0.35, we considered A=0.3 to be the best fit, due to its having the lowest mean AE.

### 4.2. The Performance of Proposed Method

Figure 5 shows an example of the $O_{2}Hb$ signal and its corresponding modulations, followed by the frequency spectrum of each modulation. It should be noted that the reference RR was 0.32 Hz.

Table 2 displays the mean AE between the reference and estimated RRs for each subject, indicating the superiority of the fusion-based strategy over the individual modulation for the majority of the subjects.

When comparing the performance of each individual modulation with the fusion-based strategy on all trials (Figure 6), the latter showed a lower mean AE (3.66) against those obtained from RAM (5.14), RFM I (4.45), RFM II (5.32), RIM I (5.61), and RIM II (5.22) BPM. According to the conducted *t*-test analysis, regarding the individual modulation, there was a significant difference between the AE of RFM I and others (p<0.05). Notably, there was also a significant difference between the fusion-based strategy and RFM I (p<0.05).

### 4.3. Comparison with the Band-Pass Filtering Method

To compare the performance of both methods, a Bland–Altman plot was used, which evaluates the agreement between two quantitative measurements by displaying their difference as a function of their average. Figure 7 shows the Bland–Altman plot of the AE values obtained by the proposed method and the method based on band-pass filtering, as used in [19] for all trials. As displayed, the proposed method outperformed the band-pass filtering with a lower average of the differences (3.66 vs. 11.06 BPM, p<0.05).

## 5. Discussion

The objective of this paper was to propose a robust RR estimation method during bicycling using fNIRS signals recorded by portable equipment. This study, by taking advantage of the lower vulnerability of fNIRS to a subject’s movements, was oriented towards highlighting the practical limitations of the current state-of-the-art studies for RR estimation during physical activity. The proposed method provides an extra measure, i.e., RR, that can be used alongside brain activity analysis. Genuinely, RR estimation from fNIRS can synergize the cerebral activity analysis in applications, such as stress assessment and mental workload, where RR changes are important indicators [35,36].

### 5.1. The Significance of Proposed Fusion Strategy

According to the literature, the performance of each extracted respiratory modulation for RR estimation depends on several factors, such as age, gender, body position, and health status [27]. Thus, a fusion-based estimation, i.e., one employing multiple respiratory modulations, can lead to a more robust RR estimation. In this paper, a fusion-based strategy was provided with an average of five respiratory modulations, based on the amplitude, intensity and frequency of an $O_{2}Hb$ signal, over a 50 s window. The comparison between the individual and fusion-based RR estimations confirmed such robustness as the 15 subjects showed lower mean AE (Table 2). Furthermore, the comparison also confirmed that the performance of an individual modulation may not be generalized for the whole population, as each modulation performed differently for different subjects. In terms of overall performance over all te trials, the fusion-based strategy outperformed each individual modulation with a lower mean of AE.

### 5.2. Comparison with State-of-the-Art

Except for our previous study, that estimated RR from an $O_{2}Hb$ signal in a resting state, there is no research estimating RR from fNIRS. Yet, a few studies have tried to portion out the respiratory components from fNIRS measurements. For instance, a multimodal extension of the general linear model using a temporal embedded canonical correlation method was employed to regress out respiratory components from fNIRS [34]. Firstly, the mentioned study required a reference respiratory signal to be recorded concurrently. Secondly, employing a general linear model usually requires a large number of fNIRS channels, which increases the complexity of wearable instrumentation. In contrast, we used a single sensor fNIRS system placed on the prefrontal cortex, which reduces the complexity of wearable instrumentation. Furthermore, the prefrontal cortex is a hairless area, and, therefore, provides the user with more comfort.

In other research, a zero delay Fourier domain band-pass filter, with cut-off frequencies from 0.2 to 0.6 Hz, was applied to the $O_{2}Hb$ signal to separate respiratory components from the cerebral brain activity [19]. The comparison between performance of the proposed method and band-pass filtering method (Figure 7) indicates the superiority of our proposed method, as a lower mean of difference between the reference and estimated RRs was obtained (3.66 vs. 11.06 BPM). The reason for such a high error could be the fact that the suggested bandwidth also includes other non-respiratory components, such as cerebral and cardiac activities.

### 5.3. Directions for Future Work

Although the results obtained by the proposed method are promising, there are some issues that should be addressed in future work. Firstly, performance of the proposed method was only investigated on young healthy subjects. It is important to consider a broader range of subjects (e.g., the elderly) to further assess the robustness of the proposed method. Furthermore, it may also be important to consider different characteristics of subjects, e.g., race, skin color, and BMI in developing an RR estimation algorithm. In particular, the employed maximum respiratory frequency ($F_{max}$) might require further adjustment for a different cohort. Secondly, although the data were recorded during physical activity and the subjects were allowed to freely move their bodies throughout the experiment, performance of the proposed method should be further validated on data where the subjects are performing more intense exercise, e.g., squatting, with more movements. Lastly, employing a smart fusion strategy, such as [37], may also further improve the performance of the proposed method. However, we did not find enormous differences amongst the estimated RRs from individual modulations (Table 2).

## 6. Conclusions

This paper presented a fusion-based strategy for RR estimation during a bicycling task from $O_{2}Hb$ signals originating from fNIRS measurements. For this purpose, five respiratory modulations were extracted from an $O_{2}Hb$ signal and, then, the RR was estimated, based on the average of dominant frequencies of each modulation. The comparison between the obtained results from the proposed method and band-pass filtering showed the superiority of the proposed method with a lower mean of AE (3.66 vs. 11.03 BPM). The importance of this study was that it addressed RR estimation during physical activity, and, by taking advantage of the motion tolerance quality of fNIRS, was oriented towards overcoming the practical limitations of traditional bio-signals, i.e., vulnerability to the subject’s movements. Furthermore, the proposed method derives an extra parameter from fNIRS signals, i.e., RR, which can be used alongside brain activity analysis.

## Figures and Tables

**Figure 1 sensors-23-03632-f001:**
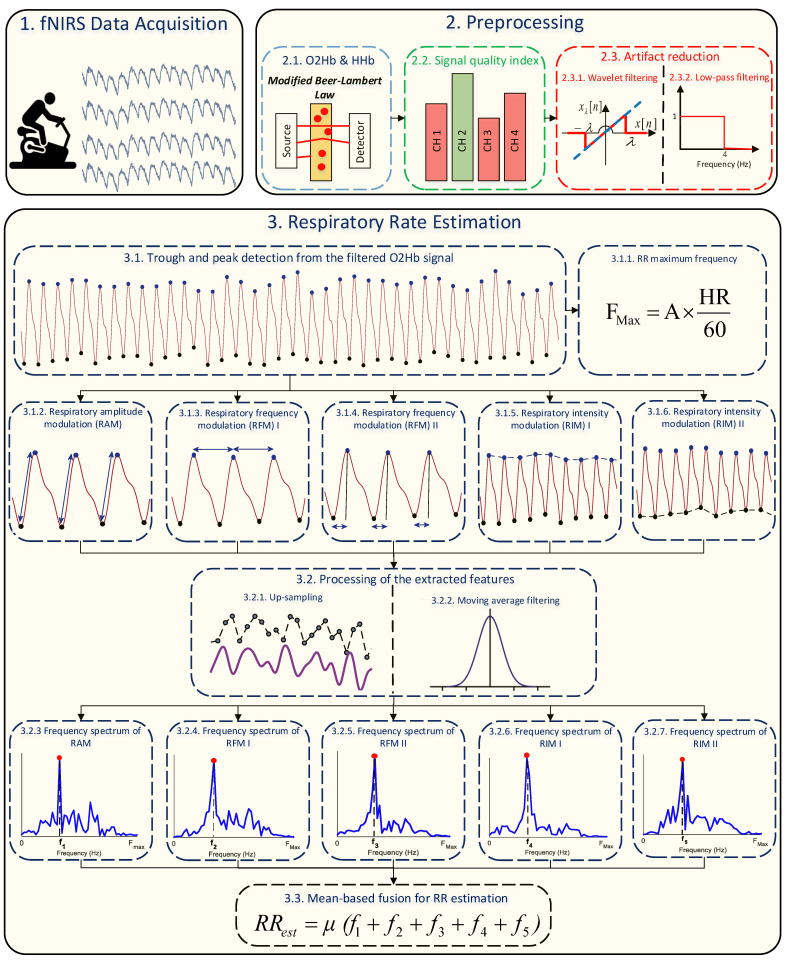
Block diagram of the proposed method for RR estimation using fNIRS.

**Figure 2 sensors-23-03632-f002:**
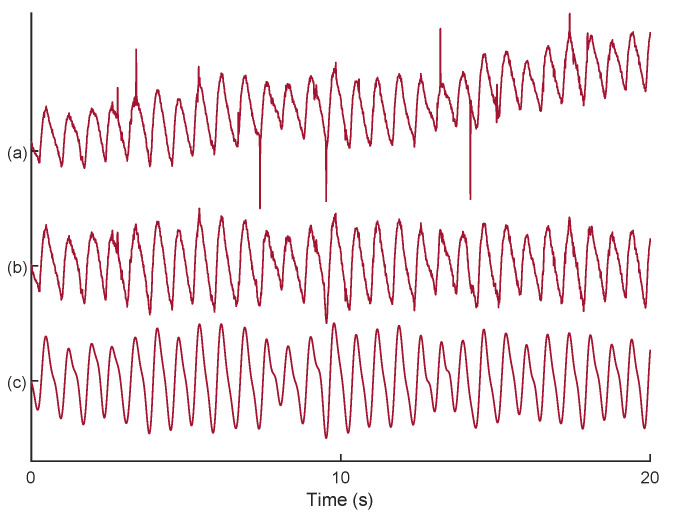
An example on noise reduction from an $O_{2}Hb$ signal: (**a**) selected $O_{2}Hb$ signal with the SQI; (**b**) motion artifact reduction using DWT algorithm; (**c**) low-pass filtering for high frequency noise removal.

**Figure 3 sensors-23-03632-f003:**
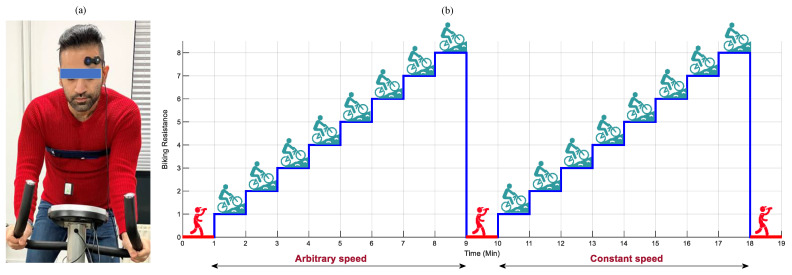
The data recording procedure. The experimental (**a**): setup and (**b**): paradigm.

**Figure 4 sensors-23-03632-f004:**
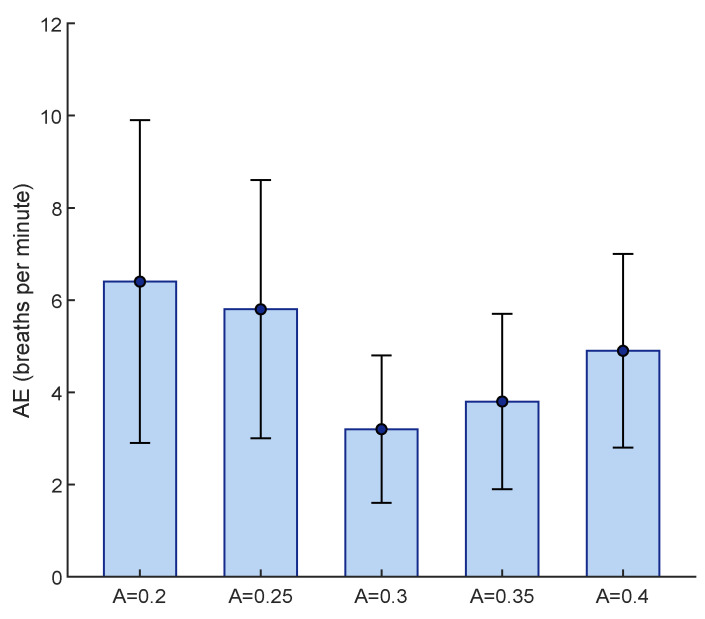
Tuning the constant A for determination of RR maximum frequency, based on lowest mean AE obtained from 78 randomly selected trials. The AE is shown in terms of mean and standard deviations.

**Figure 5 sensors-23-03632-f005:**
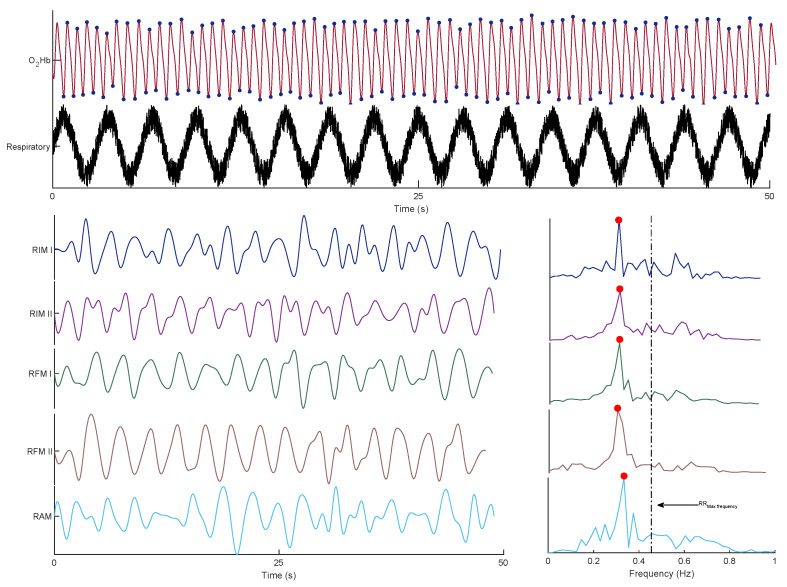
An example of the $O_{2}Hb$ signal and its corresponding respiratory signal, as well as its extracted respiratory modulations and spectra.

**Figure 6 sensors-23-03632-f006:**
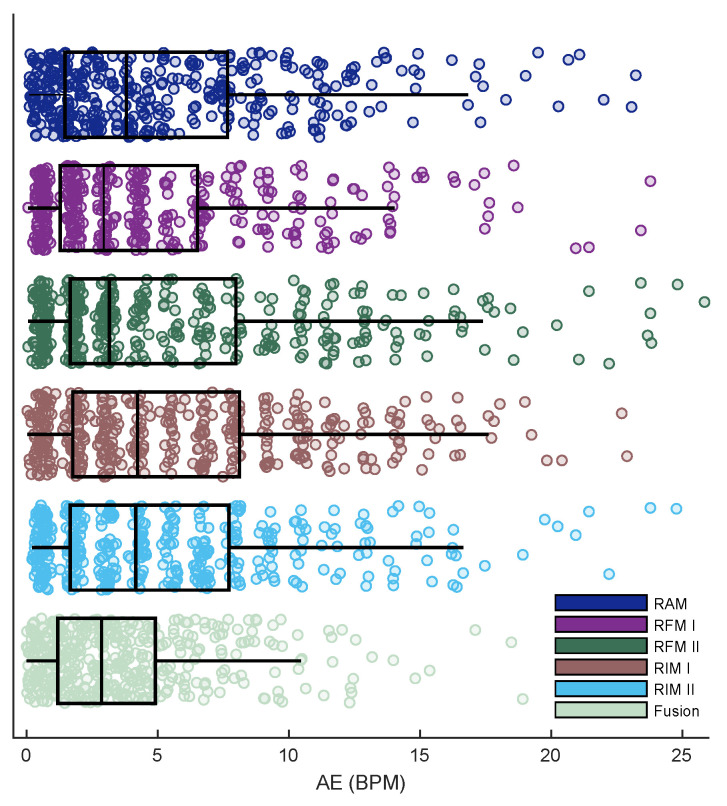
The overall performance of each individual feature and fusion-based strategy in terms of AE.

**Figure 7 sensors-23-03632-f007:**
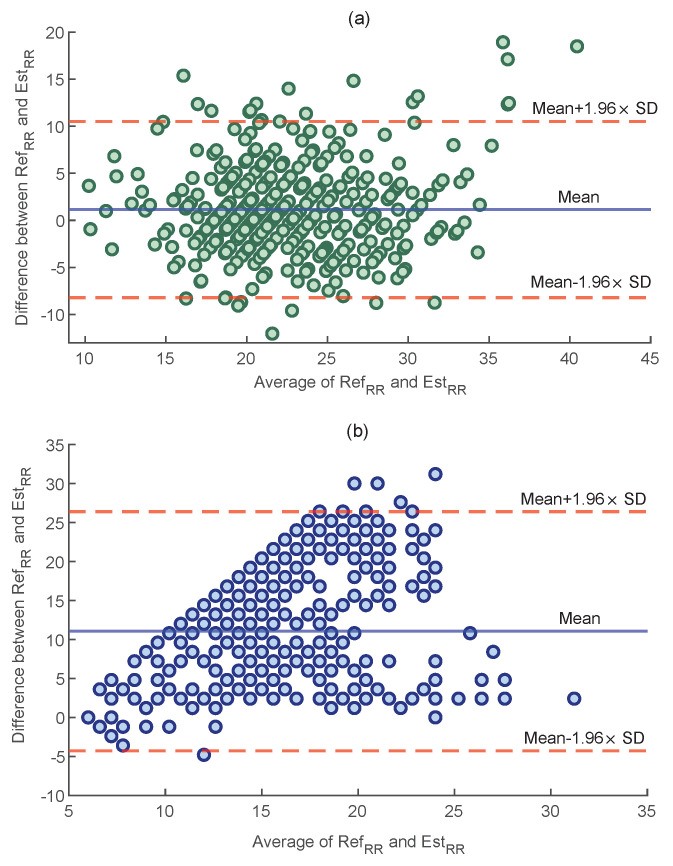
The Bland−Altman plots for: (**a**) the proposed method and (**b**) band−pass filtering [19].

**Table 1 sensors-23-03632-t001:** Characteristics of subjects who participated in the experiment. The BMI stands for the body mass index.

Subject	BMI	Age	Blood Pressure Status	Sex
1	23.4	37	Hypotension	Female
2	22.6	37	Hypotension	Male
3	20.4	24	Normotension	Male
4	18.6	33	Normotension	Female
5	25.6	22	Normotension	Male
6	19.3	22	Normotension	Male
7	19.8	26	Normotension	Male
8	21.4	31	Hypotension	Female
9	24.2	27	Hypotension	Female
10	22.3	24	Normotension	Female
11	26.8	24	Normotension	Male
12	25.4	32	Normotension	Female
13	23.8	22	Normotension	Male
14	26.2	28	Normotension	Male
15	23.4	23	Normotension	Male
16	21.8	22	Normotension	Male
17	26.7	24	Normotension	Female
18	22.9	22	Normotension	Female
19	20.8	25	Normotension	Female
20	21.5	24	Normotension	Female
21	26.4	24	Normotension	Male
22	24.6	33	Normotension	Female

**Table 2 sensors-23-03632-t002:** The mean AE between reference and estimated RRs (BPM) for each subject with individual and fusion features. The lowest mean AE is in bold.

Subject	RAM	RFM I	RFM II	RIM I	RMI II	Fusion
1	1.45	1.30	1.46	2.17	2.23	**1.05 **
2	8.54	8.72	6.30	7.79	7.17	**6.17**
3	4.85	3.72	6.91	6.27	6.9	**3.83**
4	5.62	4.01	3.27	4.95	5.23	**3.08**
5	3.29	2.53	2.75	2.53	3.25	**1.35**
6	6.48	7.78	8.50	7.60	7.77	**5.35**
7	6.51	6.76	5.15	5.49	7.02	**5.11**
8	**3.09**	3.25	4.91	5.66	3.89	3.75
9	5.96	**3.99**	10.96	9.16	8.07	5.10
10	8.66	5.38	5.18	7.47	7.68	**5.00**
11	5.98	4.31	6.21	6.93	5.43	**4.31**
12	5.47	3.61	5.01	5.05	3.9	**2.81**
13	4.97	3.09	6.09	6.47	5.31	**3.35**
14	7.62	6.82	5.63	3.98	**3.92**	4.99
15	5.87	7.30	5.62	**3.38**	5.86	3.98
16	5.73	3.67	2.88	2.54	3.00	**2.00**
17	**2.87**	3.56	5.84	8.36	6.98	3.61
18	5.34	4.07	2.40	3.73	3.84	**3.13**
19	4.73	4.03	7.26	6.98	5.09	**3.47**
20	3.71	3.69	6.85	8.59	5.31	**3.67**
21	**2.72**	2.82	4.87	4.36	3.60	2.75

## Data Availability

The data that support the findings of this study are available from the corresponding author, [science@artinis.com], upon reasonable request.

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
