# Peer review of "Estimation of Respiratory Rate during Biking with a Single Sensor Functional Near-Infrared Spectroscopy (fNIRS) System"

_sensors, 2023, doi:10.3390/s23073632_

Round 1

Reviewer 1 Report

This study aimed to estimate respiration rate during biking from fNIRS signals, namely O2Hb, with a mean-based fusion-based method. The authors present a nice article and I read it with great interest. Overall, the manuscript is well-written and the topic is beneficial for the fNIRS community and future fNIRS studies. Therefore, I would recommend the publication of this manuscript. Before that though, some aspects need to be further discussed, and some minor points need to be revised. The following are suggestions and issues that the authors should address.

·       We know that fNIRS is contaminated by other systemic physiological activities, e.g., HR and blood pressure (BP). What is the rationale that the authors focused on RR and not HR and BP? Are there any challenges to estimating HR and BP from fNIRS signals? In other words, what was the special feature/characteristic of RR, not found for example in HR, that convinced the authors to estimate this parameter (RR) from O2Hb?

·       The authors used a low-pass filter with cut-off frequency at 4 HZ to remove high-frequency noise. Since HR (~ 1.3 Hz) component can be counted as high-frequency physiological noise I was just wondering why the authors did not try to remove this component at this stage of data Pre-processing. For example, they could use a low-pass filter with cut-off frequency at 1 Hz instead of 4 Hz. The rationale should be given.

Minor:

- Line 92: “(ii) the troughs and peaks peaks”. The word “peaks” has been repeated twice.

- Line 129: In this paper, after After finding”. The word “After” should be removed.

- Line 165: “…after discarding unreliable respiratory signals, 418 50s trials”. Please rewrite the last part of this sentence. It is not clear to a reader how many trials were used for the analysis. 418 trials of the 50s each?

- Line 177: “It should be noted that the reference RR is 0.32 Hz”. Please provide a reference/references for this sentence.

Reviewer 2 Report

This paper proposes a 5 fusion-based method for estimating RR during biking from fNIRS signals recorded by a wearable 6 system where five respiratory modulations are extracted based on amplitude, frequency, 7 and intensity of the oxygenated hemoglobin concentration (O2Hb) signal. I enjoyed reading the paper and found that it is well written and well-organized. The authors have added good explanation for the methodology, and included enough results to validate their system of operation. however, the authors didn't include the differences between the different subjects who were tested, RR depends on the condition of a subject (healthy, not healthy, weight of the subject, does he have a blood pressure,etc) , the authors should include a table that shows the differences between different subjects and the different RR readings for different subjects need to be explained based on literature.  

some minor revisions need to be done before acceptance:

in section 2: proposed method 

subsection 2.1: how does the authors employ Beer lambert law in their system? 

subsection 2.3:methods under comparison:

line 2: please add citation to some of the papers you mentioned and explain their use of FNIRS and how it is different from your system and application

 remove duplicate word (after in line 6)

Results:

figure 4: what does the error bars represent ? how many data point were collected? this need to be mentioned in the text and figure caption. the figure shows an overlap between the error bars , this needs to be explained by reasons and references. 
